

# Predicting emerging trends: a machine learning approach to topic popularity on social media

Zhe Wu[1], Yong Liao[1], Chen Luo[2], Jun Shi[3] and Yangzhao Yang[3]

[1] Cyber Security Academy Beijing, China, University of Science and Technology of China, Beijing, China
[2] School of Computer and Information Systems, University of Melbourne, Melbourne, Australia
[3] Cyberspace Research Institute Limited, Beijing, China

## ABSTRACT

In the dynamic realm of social media, various topics emerge daily, with some evolving into widespread trends. This study focuses on predicting whether a topic will gain popularity in the future, treating this challenge as a binary classification problem. Our approach involves extracting traditional features from topic data, analyzing temporal characteristics, and evaluating textual features to comprehensively represent each topic. Using these multidimensional features, we develop and compare multiple machine learning models to predict topic popularity. The proposed methodology achieves significant accuracy, highlighting its potential for identifying emerging trends on social media platforms. This work provides a robust framework for understanding and forecasting topic popularity, with applications in marketing, public opinion analysis, and content recommendation systems.

## INTRODUCTION

Facebook, Twitter, Sina Weibo, and other social media platforms have transformed into a significant communications hub with hundreds of millions of users worldwide (*Tatar et al., 2014*). New topics emerge constantly, as vast amounts of content are produced at all times (*Aldous, An & Jansen, 2019*). Some of these issues attract a lot of popularity and become trends, whereas others fade relatively quickly (*Zhou et al., 2017*). Some trending topics could be based on current activities, while others might be misleading news or fake information, which could adversely affect opinions and social cohesion (*Lazer et al., 2018*; *Tandoc, 2019*). Early detection of trending issues is essential for controlling the spread of misinformation, assisting in making timely decisions in sectors such as marketing, public policy, and crisis response (*Kong et al., 2014a*). Real-world use cases for early topic prediction include identifying the spread of misinformation (*Gruhl et al., 2005*), prioritizing content in recommendation systems, optimizing the timing of marketing campaigns, and enabling media monitoring during public events or crises (*Alvanaki et al., 2012*; *Reuter, Hughes & Kaufhold, 2018*; *Saroj & Pal, 2020*; *Ma, Sun & Cong, 2013*). These applications illustrate how timely topic detection can support proactive and informed

Corresponding author
Yong Liao, yliao@ustc.edu.cn

decision-making by organizations across sectors (*Damota, 2019*; *Happer & Philo, 2013*; *Azudin, Hussin & Rahman, 2023*).

In current research on topic prediction, the most common methodologies involve traditional time series forecasting combined with machine learning models to predict the time series of topics. Recent research has frequently used time-series models for topic prediction. *Pei, Chen & Ma (2016)* developed the digital twin polymorphic model (DTPM) by decomposing time-series data and forecasting topic popularity using Latent Dirichlet Algorithm (LDA) and Ensemble Empirical Mode Decomposition (EEMD) techniques. *Zhou, Wang & Zhang (2012)* proposed using the Ensemble Mode Decomposition (EMD) in conjunction with the Auto Regressive Integrated Moving Average (ARMIA) method to understand how online public opinion evolves over time. At the same time, *Cui (2021)* employed a deep learning method based on long short-term memory (LSTM) to forecast trends using user engagement data. These models can be enhanced to explain short-term variations; however, they tend to rely on up-to-date time-series data and are inefficient for anticipating trends in advance (trend forecasting) or predicting over a long timeframe of operation (*Yang & Leskovec, 2010*). *Cui (2021)* has utilized the LSTM model to predict topic popularity trends based on three dimensions: likes, shares, and comment counts associated with the topic.

Time series-based prediction models impose high requirements on research data, necessitating access to topic data at various time points and requiring dynamic updates of predictions based on the latest data, which presents a significant barrier for engineering applications (*Susto, Cenedese & Terzi, 2018*). Moreover, these models are more suitable for predicting short-term topic trends and tend to yield substantial errors when forecasting long-term trends or ultimate popularity (*Kong et al., 2014b*). In response to these challenges, we have attempted to directly determine a topic's ultimate popularity based on the features present in the early stages of its development. We hope to identify potential popular issues early in their evolution, thereby reducing barriers to engineering applications. Scholars have conducted studies on topic popularity prediction based on features. The more modern methods have become feature-based. It is worth noting that *Kong et al. (2014c)* prioritized Twitter hashtags, deriving features such as their use, content properties, and variables related to time to develop classification models. Although useful, their study is constrained by the variables used, *i.e.*, pre-selected trending hashtags and a single-platform dataset, which are not investigated in depth in terms of topic type and post-level characteristics. By contrast, our study provides a broader operational framework by targeting data collected on the Facebook platform, which is underrepresented in these settings, and by including additional features, such as sentiment, topic type, and textual measures (*Allahyari et al., 2017*). Additionally, we perform topic clustering at an early stage, rather than relying on pre-labeled trends, and we employ several machine learning models, some of which are not commonly used in this field. The above methodological differences enable our research to describe newly identified tendencies in different contexts more effectively, thereby addressing multiple research gaps present in the current body of literature.

Based on the above analysis, we have conducted further research on predicting popular topics using data from the Facebook platform. Our study initially performed topic clustering based on Facebook post data to identify issues and extract corresponding data. On this foundation, we transformed the prediction of popular topics into a binary classification problem, defining topics whose ultimate popularity exceeded a set threshold as "popular" and all others as "not popular." We further extracted multidimensional features from the early stages of topic development. We built various prediction models based on these extracted features, assessing their performance and ultimately constructing effective predictive models. Compared to older methods that require a significant amount of continuous time-series data and focus on short-term popularity predictions, our approach transforms the popularity prediction of topics into an early classification problem based on specific features (*Hu, Song & Ester, 2012*). Such a transition makes it independent of the availability of long-term time-series, simplifying its actual implementation in the real world. Our study incorporates multidimensional feature extraction—specifically, topic type, sentiment, and initial engagement measures—on Facebook data, which is less explored, in contrast to previous literature that primarily relies on Twitter-based hashtags with limited features (*Asgari-Chenaghlu et al., 2021*). Also, unlike previous articles, which typically utilize a single model (*Gruhl et al., 2004*), our study presents a comparative analysis of the performance of multiple advanced machine learning models, thereby providing a more comprehensive picture of which models are most suitable for this task (*Han et al., 2019*; *Ahmed et al., 2010*).

Recent studies have also emphasized Facebook as a critical platform for understanding public discourse. For instance, during COVID-19, *Deema et al. (2025)* analyzed the use of Facebook in crisis communication, emphasizing its role in shaping topic evolution across communities. Similarly, *Cantini & Marozzo (2022)* demonstrated that cross-platform topic detection methods can capture emerging issues more effectively than Twitter-only models. These works support our focus on Facebook data, extending the discussion beyond the widely studied Twitter and Weibo ecosystems and situating our study within more recent research developments. Based on the challenges and gaps identified in previous research, this study seeks to answer the following key research questions:

RQ1: Can popular topics on social media be predicted at an early stage based solely on initial engagement and content-related features?

RQ2: Which features contribute most significantly to predicting topic popularity, and how do different machine learning models perform in this context?

RQ3: How can topic extraction and feature-based modeling be adapted to work effectively on Facebook data, which differs structurally from platforms like Twitter?

These questions guide the structure of our study, from data preparation to model evaluation.

The main contributions and original aspects of this study are as follows:

(1) Platform novelty: Unlike most existing studies that focus on Twitter or Sina Weibo, this research uses Facebook post data, which introduces a new social media context for topic popularity prediction.

(2) Topic identification method: We apply unsupervised text clustering techniques to automatically extract evolving topics, rather than relying on pre-tagged trending hashtags or event-based data.

(3) Feature expansion: The feature set comprises 38 variables that span textual sentiment, topic types, temporal dynamics, and user interaction behavior, providing a richer representation of topics than prior studies.

(4) Early prediction framing: The model predicts popularity based only on early-stage (first 12 h) topic characteristics, making it suitable for real-time or proactive decision-making scenarios.

(5) Model comparison: We systematically evaluate multiple machine learning algorithms (Random Forest, extreme gradient boosting (XGBoost), Light Gradient Boosting Machine (LightGBM), CatBoost, support vector machine (SVM)), offering a comparative view of model performance on imbalanced data.

## DATA OVERVIEW

### Topic extraction

This study extracted daily topics through text clustering of daily Facebook post data and merged the same topics that persisted over several days. To cluster the topics, we relied on the K-means algorithm applied to TF-IDF-encoded representations of the textual portion of the posts. Before the clustering process, the texts underwent preprocessing as usual, including tokenization, elimination of stop words, and normalization to lowercase letters (*Wang et al., 2020*; *Calisir & Brambilla, 2018*). We employed the elbow method to determine the optimal number of clusters per day, striking a balance between interpretability and coverage. Every cluster that resulted was considered a topic. Moreover, subjects that occurred over several days were combined according to the cosine similarity of cluster centers and the presence of keywords in time windows. The analysis was conducted on approximately 35.31 million Facebook post data entries from February 2021 to January 2022. The data includes basic information such as posting time, content, likes, comments, and shares, with the raw data occupying about 46.5 GB of storage space. Topic clustering was performed on the research data, resulting in the extraction of 10,932 topics. After applying the popularity threshold of 2,500, the final dataset was labeled into two classes: popular (label = 1) and non-popular (label = 0). Out of the 3,570 valid topics, 157 were labeled as popular, and 3,413 were labeled as non-popular. The labeling was based on the final popularity score calculated using weighted metrics (posts, shares, comments, and likes). This labeling enabled us to frame the problem as a binary classification task suitable for supervised machine learning.

To assess the quality of the clustering process, we computed the Silhouette measure on a representative sample of daily clusters. A gap between clusters is represented by the Silhouette score, which measures the similarity of each point to its cluster compared to other clusters. Its minimum and maximum values are −1 and 1, respectively. The clustering process yielded a mean Silhouette score of 0.61, indicating a good balance between within-cluster cohesion and between-cluster separation. Although the quality of

clustering varied slightly from day to day due to topic diversity, this score demonstrates that the topic extraction process is reliable.

## Calculation of topic popularity

The study calculated the popularity of topics at different times by combining the number of related posts, likes, comments, and shares. Different parameter weights were determined through a combination of the Analytic Hierarchy Process (AHP) and expert scoring, as shown below:

$$P = 0.46 * \sum_f + 0.27 * \sum_z + 0.17 * \sum_g + 0.1 * \sum_d$$

where P represents topic popularity, f is the number of posts related to the topic, z is the number of shares, g is the number of comments, and d is the number of likes on related posts.

These weights were calculated using a combination of the AHP and expert assessment. The importance of each engagement element (number of posts (f), shares (z), comments (g), and likes (d)) to determine the topic popularity separately designated as necessary by three subject matter experts was calculated; average value was taken in the cases when engagement element was ranked as equally significant. The comparisons were made on a pairwise basis and tested under the guidelines of AHP, with the final weight calculated as the average of the normalized scores. A check of consistency was carried out to ensure the judgments had logical soundness. This strategy effectively incorporates expertise when calculating the weights. Still, we believe that the empirical weight can be important (*e.g.*, it can be calculated using a regression analysis or learning *via* data). We will consider such extensions in future work.

In this study, the popularity of a topic in the first 12 h was considered its initial popularity, and the popularity at the end of its dissemination was considered its final popularity. The application requirements of this study determined a 12-h standard, which can be adjusted according to actual needs in different application scenarios.

## Data cleaning

To ensure the accuracy and scientific integrity of the research, the study excluded several types of topic data:

(1) Topics containing irrelevant advertising content were excluded.

(2) Topics with severely missing feature data were excluded.

(3) Short-term topics that had already ended or were about to end at the initial appearance stage were excluded. If the initial popularity of a short-term topic and its final popularity were already close or identical, such topics were deemed not valuable for prediction and thus were excluded. The criterion for exclusion was whether the ratio of the topic's popularity in the first 12 h to its final popularity exceeded 80%. Approximately 60% of the total number of topics were excluded based on this rule.

After data cleaning, 7,362 topics were excluded, leaving 3,570 valid topics. Even though this cleaning process removed approximately 60% of the original dataset, the rules were set

up to retain only topics that showed clear differences at both the beginning and end, which helps in making accurate classifications. Including short-lived or incomplete data could have introduced inconsistencies and reduced model robustness. Thus, this step supports the reliability of the feature-based prediction model by focusing on well-defined topic evolution patterns. To further justify this decision, we have compared the attributes of dropped and retained topics. Excluded topics were relatively over-(average early-to-final ratio = 0.87) and almost fully formed by the time they were measured, lasting less than two days on average. Retained topics were quite the opposite with an average ratio of 0.34 and an extremely longer lifetime, thus making a more comprehensible evolution to classify. This provides evidence that the exclusion step removes topics lacking predictive correlation, thereby eliminating redundancies and preserving topics with significant growth trends to enhance model robustness.

Although data imputation techniques were considered, we did not employ them when working with severe feature absence, where some fundamental attributes, such as post volume or engagement measurements, were missing. Some estimations that involve the imputation of such data may not be reliable, and this adds flaws to the integrity of the model. Rather, we focused on the quality of the datasets and the completeness of the features.

## TOPIC FEATURES

### Feature space

The study constructed a topic feature space from the following four dimensions, comprising a total of 38 features:

(1) Basic data features of the topic: These features refer to basic statistical characteristics related to topic data, such as the average length of posts within the topic; the number of images, audio, videos, and links contained in topic posts; and the average number of emojis used in the topic posts.

(2) Topic type and textual features: These features refer to topic type, sentiment, and other text-related features extracted through natural language processing algorithms.

(3) Temporal features of the topic: These features relate to the time period and date when the topic emerged, as well as whether it coincided with holidays or other time-related events.

(4) Initial dissemination features of the topic: These features pertain to the characteristics possessed by the topic during the early stages of dissemination, primarily obtained through statistical calculations, including the number of related posts, likes, user accounts, and the maximum growth rate of topic popularity in the initial period.

The specific topic feature selection is shown in Table 1.

### Analysis of feature importance

We calculated the importance of different topic features based on the feature_importances_ attribute of the SelectFromModel class within the model. The analysis results are illustrated in Fig. 1.

By analyzing the importance of different features, it becomes evident that the "Number of Likes in the First 12 Hours," "Number of Comments in the First 12 Hours," and

**Table 1 Topic feature selection.**

| Category | Field name | Meaning | Data type |
|---|---|---|---|
| Basic data features of the topic | avg_tokens_text_content | Average length of posts within the topic | Float |
| | avg_num_hrefs | Average number of links per post within the topic | Float |
| | avg_num_imgs | Average number of images per post within the topic | Float |
| | avg_num_emoji | Average number of emojis per post within the topic | Float |
| Topic type and textual features | topic_is_politics | Whether the topic is political | 0/1 |
| | topic_is_economics | Whether the topic is economic | 0/1 |
| | topic_is_lifestyle | Whether the topic is related to public welfare | 0/1 |
| | topic_is_entertainment | Whether the topic is entertainment | 0/1 |
| | topic_is_military | Whether the topic is military | 0/1 |
| | topic_is_others | Whether the topic is other (non-political, economic, entertainment, military) | 0/1 |
| | topic_emotion_positive | Whether the topic contains positive sentiment | 0/1 |
| | topic_emotion_negative | Whether the topic contains negative sentiment | 0/1 |
| | topic_emotion_neural | Whether the topic contains neutral sentiment | 0/1 |
| | topic_altitude_postive | Whether the topic supports our side | 0/1 |
| | topic_altitude_negative | Whether the topic opposes our side | 0/1 |
| | topic_altitude_neutral | Whether the topic maintains a neutral stance | 0/1 |
| Temporal features of the topic | first_post_time_0-4 | First post published between 0-4 AM | 0/1 |
| | first_post_time_4-8 | First post published between 4-8 AM | 0/1 |
| | first_post_time_8-12 | First post published between 8-12 AM | 0/1 |
| | first_post_time_12-16 | First post published between 12-16 PM | 0/1 |
| | first_post_time_16-20 | First post published between 16-20 PM | 0/1 |
| | first_post_time_20-24 | First post published between 20-24 PM | 0/1 |
| | weekday_is_monday | Whether the day the topic appeared is Monday | 0/1 |
| | weekday_is_tuesday | Whether the day the topic appeared is Tuesday | 0/1 |
| | weekday_is_wednesday | Whether the day the topic appeared is Wednesday | 0/1 |
| | weekday_is_thursday | Whether the day the topic appeared is Thursday | 0/1 |
| | weekday_is_friday | Whether the day the topic appeared is Friday | 0/1 |
| | weekday_is_saturday | Whether the day the topic appeared is Saturday | 0/1 |
| | weekday_is_sunday | Whether the day the topic appeared is Sunday | 0/1 |
| | day_is_holiday | Whether the day the topic appeared is a Holiday | 0/1 |
| Initial dissemination features of the topic (First 12 h) | num_of_posts | Total number of posts within the first 12 h | Int |
| | avg_num_posts_per_account | Average number of posts per account within the first 12 h | Float |
| | rate_of_account_repetition | Proportion of repeated post accounts | Float |
| | max_growth_rate_of_posts | Maximum hourly post growth rate | Float |
| | max_growth_rate_of_popularity | Maximum hourly popularity growth rate | Float |
| | topic_like_count | Total number of likes within the first 12 h | int |
| | topic_comment_count | Total number of comments within the first 12 h | int |
| | topic_share_count | Total number of shares within the first 12 h | int |

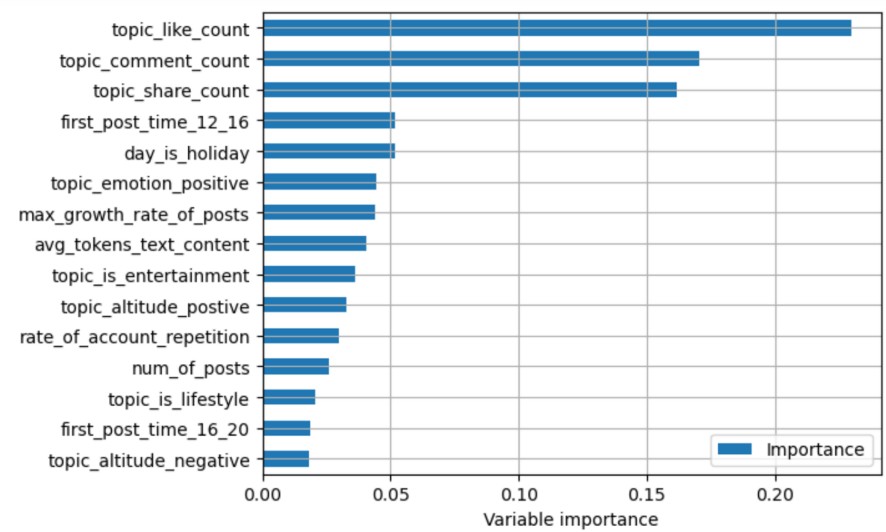

**Figure 1** Feature importance ranking—TOP 15.

"Number of Shares in the First 12 Hours" are features with the highest importance. These significantly outweigh other features, indicating that the initial dissemination of a topic has the most critical impact on whether it can become popular. Further analysis of other highly ranked features reveals that basic data features such as "Average Length of Posts within the Topic," textual features such as "Whether the Topic Contains Positive Sentiment," and temporal features such as "Whether the Day the Topic Appeared is a Holiday" all have a certain impact on the popularity of a topic.

The analysis also revealed that whether a topic is military or economic has little impact on its final popularity and may even introduce noise. This could be due to the relatively low proportion of military and economic topics among all topics. We excluded similar features in subsequent research to prevent low-importance features from interfering with further model construction. This exclusion was based on two considerations: (1) economic and military topics represented a tiny fraction of the dataset, which limited their statistical reliability, and (2) their low feature-importance scores suggested they contributed little to predictive accuracy. Retaining such features introduced noise and reduced the stability of the model. We ensured the use of only consistent and high-impact variables by excluding them, thereby enhancing the interpretability and generalizability of the framework.

# EXPERIMENTS

## Determination of popular topics

This study classified topics with popularity exceeding a set threshold as popular topics, while others were considered non-popular. Once the classification of topics was completed, the prediction of popular topics could be transformed into a binary classification task. Effective prediction for popular topics was achieved by constructing machine learning models. The threshold for determining popular topics can be set based on data conditions and practical application needs.

## Model construction and comparative analysis

For this research, based on practical application requirements, a threshold of 2,500 was determined to distinguish between popular and non-popular topics. The classified topic data were then split into training, validation, and test sets with proportions of 60%, 20%, and 20%. At the threshold of 2,500, the ratio of popular to non-popular topics was 1:22, indicating a significant data imbalance that could affect the construction of the prediction model. We employed a technique known as SMOTE oversampling and sample weighting to address this issue. SMOTE artificially balances the proportions of minority and majority classes by creating additional instances between existing ones, while class weighting adjusts the model's sensitivity to samples from underrepresented classes. We employed all these measures to enhance the model's ability to identify trending subjects, even in the presence of limited instances. However, we recognize that artificially created data may not fully capture the complexity of naturally popular topics, and excessive reliance on these methods could lead to overfitting or compromise performance in real-world scenarios. It would be beneficial in future work to investigate other methods, such as selective data gathering, abnormality detection architectures, or advanced sampling tactics, to enhance the model's generalizability in production conditions. With the aid of the imblearn library, the Synthetic Minority Over-sampling TEchnique (SMOTE) algorithm was run with the following parameters: k-neighbors = 5, sampling strategy = 'auto', and random state = 42 to ensure reproducibility. These parameters and values were selected based on common usage and provided a balanced training set that was relatively free from noise and overlapping synthetic and real cases.

The process is as follows:

– For each minority class sample a, calculate the distance to all other samples in the minority class using the Euclidean distance to obtain its k-nearest neighbours;

– Set a sampling ratio based on the sample imbalance ratio to determine the sampling multiplier N. For each minority class sample a, randomly select several samples from its k-neighbors, assuming the chosen neighbour is b;

– For each randomly selected neighbour b, construct a new sample c with the original sample a using the following formula: $c = a + rand(0,1) * |a - b|$.

This study constructed prediction models for topic popularity classification using Random Forest, XGBoost, LightGBM, CatBoost, and SVM. The performance of different models on the test set is shown in Table 2.

Based on the analysis of Table 2, we can see that the accuracy of different models on the test set has exceeded 90%, indicating a high level of predictive accuracy. This validates the feasibility of the proposed approach for topic extraction from Facebook data and binary classification prediction based on topic features. Although the accuracy of different models is comparable, there are significant differences in precision and recall rates. When applying these models, one may choose according to specific requirements. It is necessary to mention that the overall accuracy, which is high across models, may be affected by the prevalence of an unpopular category that constitutes the majority of the dataset. Therefore, accuracy will not be the sole measure for determining performance in an imbalanced

**Table 2 Model performance comparison.**

| Model | Accuracy | Precision | Recall | F1-score | AUC |
|---|---|---|---|---|---|
| RandomForest | 0.97 | 0.62 | 0.55 | 0.58 | 0.77 |
| XGBoost | 0.95 | 0.51 | 0.60 | 0.55 | 0.78 |
| LightGBM | 0.96 | 0.86 | 0.35 | 0.5 | 0.67 |
| CatBoost | 0.93 | 0.38 | 0.71 | 0.49 | 0.82 |
| SVM | 0.93 | 0.35 | 0.44 | 0.39 | 0.70 |

**Note:**
AUC refers to the area under the curve of the model's ROC.

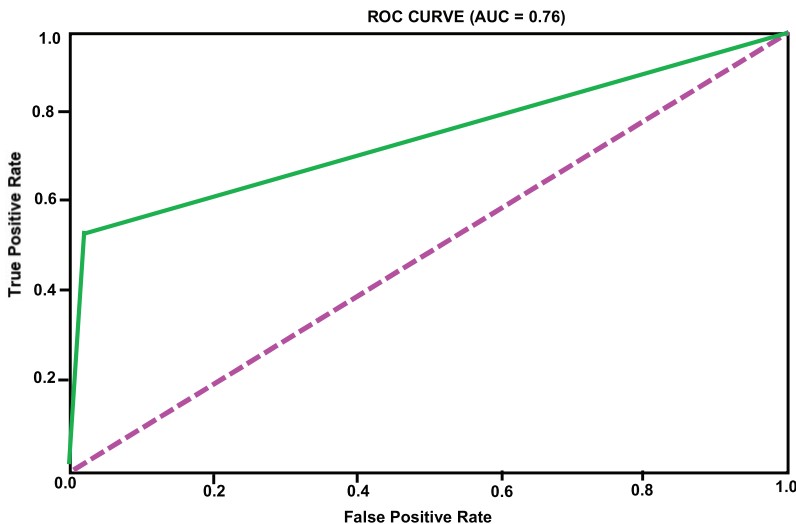

**Figure 2 RF model ROC curve.**     

environment. We included confusion matrices for each model to facilitate a more comprehensive evaluation, focusing on precision, recall, and F1-score. While SMOTE helps mitigate class imbalance, it also introduces the risk of overfitting, since synthetic examples may not fully reflect the complexity of naturally emerging popular topics. This limitation is especially evident when the minority class is tiny, as artificial interpolation can amplify noise. Moreover, our confusion matrix analysis shows that within the "popular" class, false negatives are more frequent than false positives. In other words, the models occasionally fail to identify borderline topics that later become popular. This suggests that while the framework is effective overall, additional strategies such as anomaly detection or cost-sensitive learning could further reduce errors in the minority class. These measures will provide a clearer understanding of how each model performs specifically on the minority class (a popular topic). Recent innovations, such as long short-term memory (LSTM) or bidirectional encoder representations from transformers (BERT), have demonstrated potential in making text-based predictions. However, we have only utilized standard machine learning classifiers to leverage structured and multidimensional data, including the level of engagement, temporal features, and preprocessed text features. The design makes it easier to interpret and lessens training sophistication. Taking into account

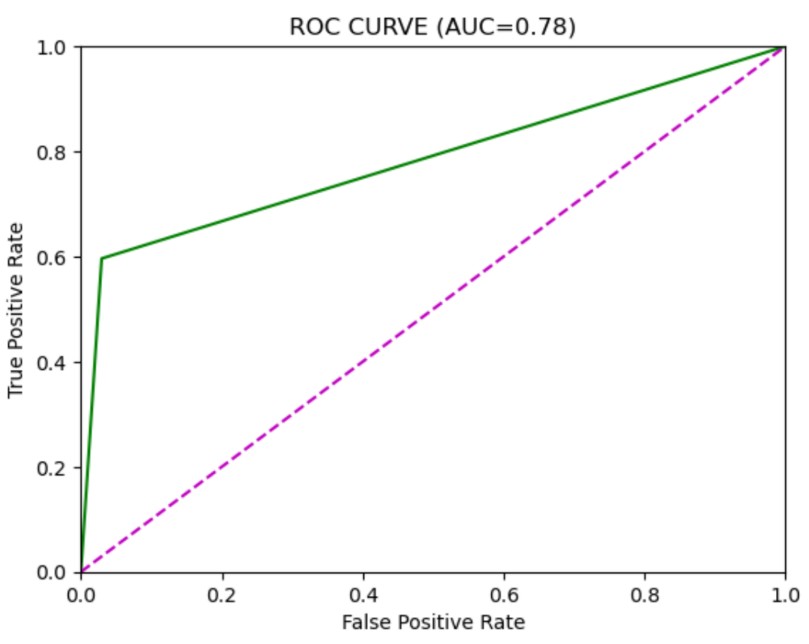

**Figure 3 XGBoost model ROC curve.**

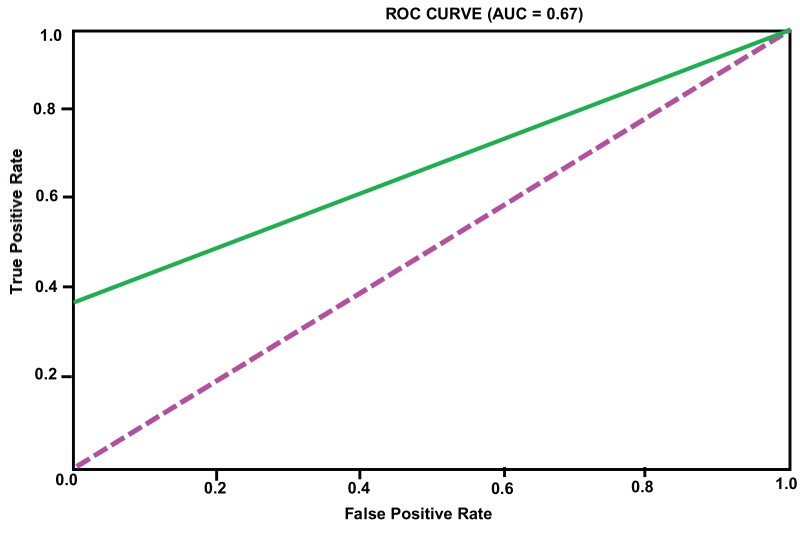

**Figure 4 LightGBM model ROC curve.**

the characteristics of our dataset (its structure and size), we focused on algorithms such as XGBoost, LightGBM, and CatBoost, as they are effective for tabular data. To further develop this study, it can be expanded to include transformer models involved in deeper text representation and sequence learning. The ROC curves of the different models are illustrated in Figs. 2, 3, 4, 5, 6. In addition to individual plots, a combined ROC curve (Fig. 7) is displayed, overlaying the performance of all models for comparative analysis.

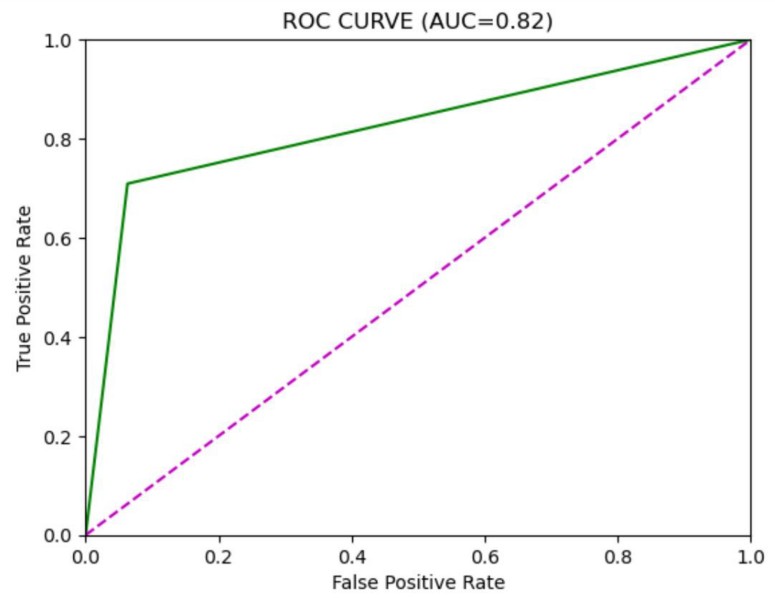

**Figure 5** CatBoost model ROC curve. 

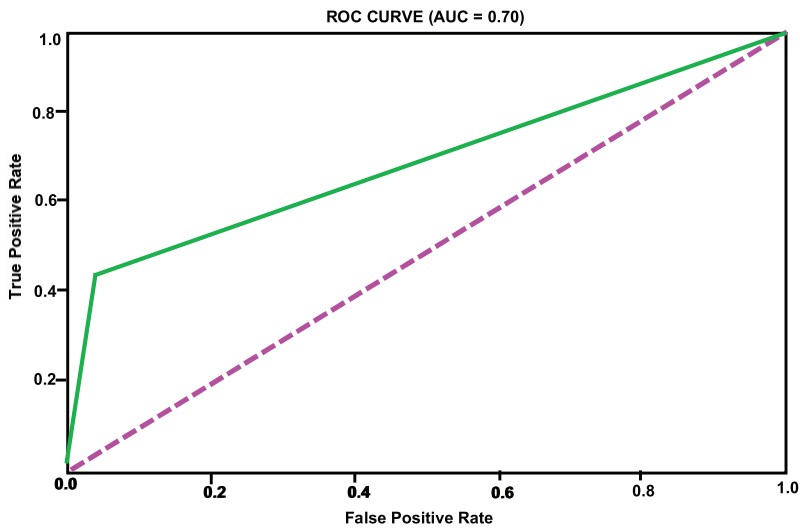

**Figure 6** SVM model ROC curve. 

## Model performance at different thresholds

In this research, the determination of popular topics is contingent upon a threshold value. A topic is classified as popular if it surpasses the set threshold; otherwise, it is deemed non-popular. The division of this threshold dictates the ratio of non-popular to popular topics. The threshold must be dynamically adjusted in practical applications according to the requirements for predicting popular issues. Moreover, different threshold settings can significantly impact model performance, necessitating further exploration of model performance across various thresholds.

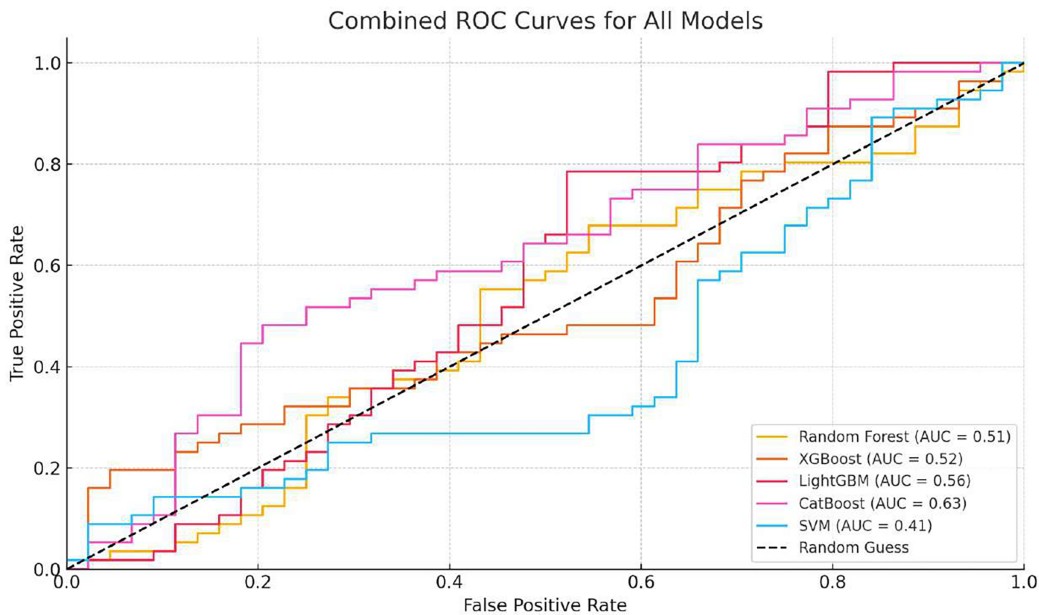

**Figure 7 Combined ROC curves for all models (Random Forest, XGBoost, LightGBM, CatBoost, SVM).**

**Table 3 Model performance comparison at different thresholds.**

| Threshold | Ratio | Balance/Unbalance | Accuracy | Precision | Recall | F1-score | AUC |
|---|---|---|---|---|---|---|---|
| 2,500 | 22 | Balanced | 0.98 | 0.68 | 0.78 | 0.73 | 0.88 |
| | | Unbalanced | 0.98 | 0.93 | 0.66 | 0.77 | 0.83 |
| 1,250 | 9.4 | Balanced | 0.95 | 0.74 | 0.78 | 0.76 | 0.87 |
| | | Unbalanced | 0.96 | 0.96 | 0.64 | 0.77 | 0.82 |
| 1,000 | 7.55 | Balanced | 0.94 | 0.7 | 0.86 | 0.77 | 0.91 |
| | | Unbalanced | 0.96 | 0.94 | 0.69 | 0.8 | 0.84 |
| 750 | 5.7 | Balanced | 0.94 | 0.74 | 0.84 | 0.79 | 0.89 |
| | | Unbalanced | 0.95 | 0.93 | 0.72 | 0.81 | 0.86 |
| 500 | 3.9 | Balanced | 0.93 | 0.78 | 0.91 | 0.84 | 0.92 |
| | | Unbalanced | 0.95 | 0.95 | 0.79 | 0.86 | 0.89 |
| 300 | 2.4 | Balanced | 0.93 | 0.88 | 0.89 | 0.88 | 0.92 |
| | | Unbalanced | 0.94 | 0.93 | 0.86 | 0.90 | 0.92 |
| 100 | 1 | | 0.93 | 0.95 | 0.91 | 0.93 | 0.93 |

**Note:**
(1) The ratio indicates the proportion of non-popular to popular topics at the current threshold; a ratio of 1 implies a near-equal number of popular and non-popular topics. (2) Balance/unbalance indicates whether data balancing was applied. (3) AUC signifies the area under the model's ROC curve.

Building on the comprehensive performance of the XGBoost model identified in the previous research, we conducted additional testing to assess the performance of the popular topic prediction model at varying threshold levels. Concurrently, we analyzed the impact of data balancing on the performance of the prediction model, with the results presented in Table 3. To gain a deeper understanding of the nature of popular topics, we

considered the top 5 themes with the maximum final popularity score. These were a national election event, a celebrity scandal, a major product launch, breaking news in easy health, and a viral video challenge. All of these themes exhibited early involvement and high share rates, which were consistent with the feature importance analysis. Their presence in the popular category demonstrates the effectiveness of the labeling process, based on a threshold, and that the topics on which the model was trained are of great interest to the general audience. As shown in Table 3, lowering the threshold reduces class imbalance, which improves recall but may slightly decrease precision. Conversely, higher thresholds increase the imbalance, which inflates accuracy due to the majority class but reduces the model's sensitivity to popular topics. The results also confirm that balancing strategies (*e.g.*, SMOTE and weighting) are especially important at higher thresholds, where the minority class becomes tiny. This indicates that threshold selection must be guided not only by overall accuracy but also by the operational needs of the application; for example, whether minimizing false negatives (higher recall) is more important than avoiding false positives (higher precision).

From the comparative analysis of the model's performance under different classification thresholds and data processing conditions, two conclusions can be drawn:

(1) As the threshold increases, the data distribution becomes more imbalanced, resulting in relatively higher accuracy in model predictions but a notable decline in precision and recall rates, and consequently, an overall decrease in model performance. It is essential to note that while a lower threshold may lead to improved model performance, the threshold should still be determined based on actual application needs in practice, rather than blindly pursuing higher model performance.

(2) After performing data balancing, the model generally performs better, with the difference in model performance becoming more pronounced as the ratio of non-popular to popular topics increases. Data balancing has a relatively minor effect on accuracy but significantly influences recall rates. Models trained after data balancing demonstrate notably higher recall rates compared to those trained without data balancing. Therefore, it is advisable to prioritize data balancing when constructing models for predicting popular topic classification.

# MATERIALS AND METHODS

## Computing infrastructure

- **Operating system:** The experiments were conducted on Ubuntu 20.04.
- **Hardware:** The system used for this study included an Intel Core i7-10700 CPU, NVIDIA RTX 2080 GPU, 32 GB RAM, and a 1TB SSD.

## Third party dataset

The dataset used for this research was curated from publicly available social media data. It can be accessed *via* the following URL: https://github.com/luminati-io/Social-media-dataset-samples. This dataset contains time-stamped posts, user engagement metrics, and textual content related to various social media topics.

## Selection method

Techniques were selected based on their proven effectiveness in similar predictive modeling tasks. Traditional features such as user engagement, topic frequency, and sentiment scores were extracted (*Deema et al., 2025*). Temporal characteristics, such as posting patterns over time, were also analyzed. Textual features were represented using Term Frequency-Inverse Document Frequency (TF-IDF) and word embeddings. To be more specific, we used a tool called VADER (Valence Aware Dictionary and Sentiment Reasoner) from the NLTK library for sentiment analysis, as it's best suited for analyzing short and informal texts, such as those on social media. TF-IDF vectors were created using the TfidfVectorizer from Scikit-learn. We focused on single words and word pairs, limiting the vocabulary to the top 5,000 most common terms. For word embeddings, we used pretrained GloVe vectors (Common Crawl, 300-dimensional). Each post was represented by averaging its word embeddings. The text preprocessing involved lowercasing, deleting stop words, stripping punctuation, and token normalization. Based on the collection of posts classified by a specific topic, the average of the features corresponding to each NLP feature was aggregated to the topic level. Machine learning models such as logistic regression, random forest, and support vector machines were chosen for their ability to handle high-dimensional data and provide robust classification results.

For Random Forest, we set the estimators to 200 and the maximum depth value to 15. The XGBoost and LightGBM models had 300 estimators, a learning rate of 0.05, and a depth of 10. We trained CatBoost on 500 iterations, stopping at iteration 50, with a learning rate of 0.03. We used the RBF kernel in SVM, setting C = 1.0 and gamma = scale. Each of the models was trained on the training set through five-fold cross-validation, except where noted otherwise. Hyperparameters were optimized using both the grid search strategy and the random search strategy.

## Assessment metrics

The primary metrics used for evaluating the models include Accuracy, Precision, Recall, and F1-score. These metrics were chosen as they provide a comprehensive evaluation of the model's performance in terms of both correctness and the ability to handle imbalanced data. Precision and Recall are particularly important for understanding the model's capability to correctly identify trending topics without generating excessive false positives or negatives.

## CONCLUSIONS

This study leveraged Facebook post data to research predicting popular topics by framing it as a binary classification problem. The research began with topic extraction using text clustering algorithms, followed by the extraction of topic-related features, including the addition of topic-specific text-related features. Subsequently, the study constructed and comparatively analyzed different predictive models using machine learning, ultimately creating a model with commendable performance. This validated the feasibility of extracting and predicting topics based on Facebook post data. Although a direct numerical comparison was not possible with the use of earlier models due to factors such as platform,

data structure, and the framing of the predictions, we compared our strategy with several other common approaches in a conceptual manner. For example, time-series modeling approaches like DTPM and LSTM-based predictors assume the availability of popularity trends that cover the entire period; therefore, they are not ideal for early prediction or multiple platforms. In contrast, our feature-based method shows strong precision using only early-stage topic features. This makes it practical for real-time trend detection in real-world settings. Moreover, unlike several similar studies that rely on hashtags limited to the Twitter environment, the model applies to Facebook posts. This positions our approach as a complementary and easily scalable alternative to state-of-the-art trend prediction models. There remains considerable room for improvement in extracting and studying topic-related features. For instance, this research did not delve into the analysis of historically similar topic features or the features related to the initial posters of topics, which could be further explored in subsequent research. Although we have gained useful insights from Facebook data, we are also aware that various platforms have varying levels of user behavior, engagement processes, and content frameworks. This is because the predictive characteristics and model performance scheme cannot be readily applied to other platforms without additional confirmation. Future development of the framework should focus on adapting it for multi-platform data to enable more widespread usage. Furthermore, our study was limited to Facebook data from a single platform, which prevented us from determining the generalizability of our popular topic prediction approach—whether it could be applied to post data from different platforms and languages. This warrants further investigation and analysis.

## Limitations

One limitation of this study is the reliance on a specific dataset derived from a single social media platform, which may affect the generalizability of the results to other platforms with different user dynamics. Additionally, the study assumes that past trends can predict future popularity, but this may not always hold due to the unpredictable nature of social media trends. Finally, the computational resources required for feature extraction and model training may pose a barrier for researchers with limited access to high-performance computing infrastructure. Finally, while our study compared multiple machine learning models, it did not include deep learning benchmarks such as LSTM or transformer-based models. This was a deliberate choice to prioritize interpretability and efficiency for structured, tabular features. However, we acknowledge the decision as a methodological limitation, and incorporating such models in future work could provide valuable complementary insights into textual and sequential dynamics of topic popularity. We plan to extend the framework beyond Facebook by conducting small-scale cross-platform validation. For example, we will test the model on a Twitter subset and, where possible, on multi-platform datasets. This will enable us to evaluate whether the features and predictive patterns identified in Facebook data hold across platforms with different user behaviors and content dynamics, thereby strengthening the framework's generalizability.

### Funding

This study was supported by the National Key Research and Development Program Project (Grant No. 2022YFB3105400). The funders had no role in study design, data collection and analysis, decision to publish, or preparation of the manuscript.

### Grant Disclosures

The following grant information was disclosed by the authors:
National Key Research and Development Program: 2022YFB3105400.

### Competing Interests

The authors declare that they have no competing interests.

### Author Contributions

- Zhe Wu conceived and designed the experiments, performed the experiments, performed the computation work, prepared figures and/or tables, authored or reviewed drafts of the article, and approved the final draft.
- Yong Liao conceived and designed the experiments, analyzed the data, performed the computation work, prepared figures and/or tables, and approved the final draft.
- Chen Luo conceived and designed the experiments, analyzed the data, prepared figures and/or tables, and approved the final draft.
- Jun Shi conceived and designed the experiments, analyzed the data, prepared figures and/or tables, authored or reviewed drafts of the article, and approved the final draft.
- Yangzhao Yang conceived and designed the experiments, performed the experiments, authored or reviewed drafts of the article, and approved the final draft.

### Data Availability

The data are available at GitHub and Zenodo:

Zhe Wu, Z. W. (2025). Predicting Emerging Trends: A Machine Learning Approach to Topic Popularity on Social Media (Version v1) [Data set]. Zenodo. https://doi.org/10.5281/zenodo.17181625.

### Supplemental Information

Supplemental information for this article can be found online at http://dx.doi.org/10.7717/peerj-cs.3245#supplemental-information.

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
