# Peer review of "Predicting emerging trends: a machine learning approach to topic popularity on social media"

_PeerJ Computer Science, doi:10.7717/peerj-cs.3245_

## Round 0.1 · original submission · Major Revisions

· Academic Editor

Major Revisions

Dear Authors,

Thank you for submitting your article. Based on the reviewers' comments, your article has not yet been recommended for publication in its current form. However, we encourage you to clearly address the concerns and criticisms of the reviewers and to resubmit your article once you have updated it accordingly. Reviewer 2 has suggested you to provide specific references. You are welcome to add them if you think they are useful land relevant. However, you are under no obligation to include them, and if you do not, it will not affect my decision.

Best wishes,

**Language Note:** The review process has identified that the English language must be improved. PeerJ can provide language editing services - please contact us at [email protected] for pricing (be sure to provide your manuscript number and title). Alternatively, you should make your own arrangements to improve the language quality and provide details in your response letter. – PeerJ Staff

Reviewer 1 ·

Basic reporting

Although the article is written in a generally understandable language, inconsistencies and irregularities in expression are noticeable in some sections. Especially in the sections where technical terms are explained, the sentences are too long and complicated. Therefore, the study needs a comprehensive language and expression arrangement.
The literature review remains superficial and does not reflect the depth of the subject. Existing studies are passed through with short references, but the gaps that the study fills and the aspects in which it differs from previous methods are not sufficiently discussed. There are references to methods commonly used in the literature, but a comparative evaluation of the current approach with these methods is not presented. Authors should strengthen the introduction and literature sections in a way that clearly demonstrates the original contribution of the study.
In addition, tables and figures are limited to the presentation of only quantitative data; interpretative explanations that integrate visual elements with the analysis and support the discussion are missing. Figure2-Figure6 (ROC Curve) can be combined into a single image. The explanations placed under the tables should be made more technical and explanatory.

Experimental design

Although the research design seems detailed at first glance, some basic methodological explanations are insufficient, which negatively affects the reproducibility of the study. In particular, the details of the natural language processing methods used in the feature engineering process are unclear. For example, technical information such as which sentiment analysis tools are used, how TF-IDF or embedded representations are implemented are not clearly stated. This deficiency makes the technical accuracy of the study questionable.

The data belongs only to the Facebook platform, which seriously limits the generalizability of the findings of the study. The authors touch on the dynamics of multiple social media platforms but do not test their own models in this context. This raises questions about the practical validity of the study. In addition, the exclusion of 60% of the data weakens the representativeness of the analyzed sample; this makes the reliability of the study questionable.

Although the classification algorithms used in the modeling process are up-to-date and appropriate, it is not explained why more advanced artificial intelligence methods (e.g. LSTM, BERT or transformer-based models) were not preferred. Considering the data structure and volume, the applicability of these models should have been investigated. Additionally, no technical details regarding model hyperparameters are provided, which undermines the scientific transparency of the study.

Validity of the findings

The findings presented in the study seem to be consistent with the methods used, but there are serious concerns about the validity of the findings. Although the imbalance between classes is very evident, trying to overcome this problem only by using SMOTE raises doubts about whether the model is learning healthily. In particular, the bias caused by the scarcity of examples belonging to the "popular" class may cause the model to exhibit low performance on real data that it may encounter in the field.

Although the model performance metrics appear high, it has not been fully established to what extent these metrics are affected by the class imbalance. For example, it is understood that the reason behind the high accuracy rates is most likely the weight of the "non-popular" class. In such a data imbalance situation, accuracy alone is not a meaningful indicator; however, this situation has not been discussed sufficiently in the article. Cofusion matrices should be added for a more detailed evaluation.

In addition, the limitations stated at the end of the study remain quite superficial and cliché. Expressions such as "different platforms" are used; however, no concrete methodological solution suggestions are presented regarding these limitations. It would have been expected that the suggestions for future studies would be more analytical and original. It is also important for scientific integrity that the authors discuss under what conditions their results may be invalid, but such questioning is not included in the text.

·

Basic reporting

Overall the research work is related to trending research finding using Machine Learning. This is good research area. Overall topic is good. However a number of comments need the authors attention as given in additional comments

Experimental design

How the data is prepared and labeled is the main concern.

Validity of the findings

The results computed are good. However how these are validated against the existing studies?

Detailed analysis of the paper is presented in additional comments.

Additional comments

In this research study in the authors have focused on predicting the popularity of various research topics using machine learning. This is a unique and different idea that the topic which are popular on social media will be classified using learning algorithms. Study is good however there are many issues and comments that need the authors attention:
1) data lebelling is the main concern. How the data is labelled? how many classes are there? if there are only two classes which is popular yes or not then how many number of topics are there which are labelled and how they were labeled? . The list of the total number of topics as well as the topic related results should be presented in the dataset details section and also the top trending topics should be highlighted and disccused in the results section.
2) For topic popularity an equation is given, however Logic behind Feature Significance Which is shown with the help of Constant numbers is not discussed. For instance 0.46 is considered for a weight of the first feature, how it is calculated? . The weight of the features should be calculated based on empirical analysis.
3) ROC curves are given separately these should be given in one hours to show the comparative analysis.
4) it is expected that the results of the proposed approach be compared with the existing studies which is the main concerns and no such compared then comparison with the state of Art of the proposed approach should be given.
5) the existing studies are not explored and compared so number of research studies are here whcih should be presented in the related work, few are as follows:

Zou, T., Guo, P., Li, F., & Wu, Q. (2024). Research topic identification and trend prediction of China's energy policy: A combined LDA-ARIMA approach. Renewable Energy, 220, 119619.

Rodrigues, A. P., Fernandes, R., Bhandary, A., Shenoy, A. C., Shetty, A., & Anisha, M. (2021). Real‐Time Twitter Trend Analysis Using Big Data Analytics and Machine Learning Techniques. Wireless Communications and Mobile Computing, 2021(1), 3920325.

Zhu, E., et al (2024). PHEE: Identifying influential nodes in social networks with a phased evaluation-enhanced search. Neurocomputing, 572, 127195. doi: https://doi.org/10.1016/j.neucom.2023.127195

Peng, Y., et al. (2024). Unveiling user identity across social media: a novel unsupervised gradient semantic model for accurate and efficient user alignment. Complex & Intelligent Systems, 11(1), 24. doi: 10.1007/s40747-024-01626-6

Khan, H. U., et al. (2021). Twitter trends: a ranking algorithm analysis on real time data. Expert Systems with Applications, 164, 113990.

Reviewer 3 ·

Basic reporting

The studies in the literature were not investigated in detail (only 3 studies were given) and an adequate literature review was not presented. In addition, no critical analysis was made of the relevant studies and the strengths/weaknesses of the studies were not discussed. The superiority of the proposed method over these studies was not emphasized.

The original aspects of the study should be emphasized more clearly. The listed contributions are given in general. It should be written in a way that reveals the originality of the study with concrete expressions and details.

The research questions sought to be answered regarding the reason for this study and the early detection of popular topics should be clearly stated.

Experimental design

In the Topic Extraction section, it is stated that text clustering was used, but the clustering method used and its details are not given.

It is stated that 10,932 topics were extracted, but the quality of these clusters was not evaluated with metrics such as the Silhouette score.

In the calculation of the topic popularity section, it is stated that the weights were calculated with a combination of the Analytic Hierarchy Process and expert scoring. However, the process of determining the weights is unclear. Details such as the implementation method of the Analytic Hierarchy Process, the number of experts, areas of expertise, etc. should be provided.

Topics where feature data is seriously lacking have been excluded. Couldn't imputation methods be used?

In determining popular topics, the threshold value was selected as 2500 according to practical application needs. This threshold selection process should be detailed.

SMOTE algorithm parameters should be provided.

RandomForest, XGBoost, LightGBM, CatBoost and SVM methods were used, but sufficient information about the implementation of these methods was not provided.

Figures and tables should be explained in more detail and the performances of the methods should be discussed.

Validity of the findings

The application validity of the proposed study is limited. It is not stated in which real-world scenarios it can be used.

The feature set used is limited to only the post content.

---

## Round 0.2 · Minor Revisions

· Academic Editor

Minor Revisions

Dear Authors,

Although two reviewers recommend accepting your paper, one reviewer suggests minor revision. We encourage you to address the concerns and criticisms of Reviewer 1 and resubmit your paper once you have updated it accordingly.

Best wishes,

Reviewer 1 ·

Basic reporting

Language and Expression: The authors state that they have simplified long and complex sentences, and the revised manuscript indeed demonstrates more fluent language. However, in the Methodology section (e.g., lines 340–373), some sentences still contain excessive technical clustering. Shorter sentences and clearer sub-sectioning are recommended.

Literature Review: The review has been expanded and deepened, and the ways in which the study diverges from previous methods are now more clearly articulated. Nevertheless, the literature section remains heavily focused on references related to “Twitter and Sina Weibo.” The inclusion of more recent references (2023–2025) that strengthen the Facebook context would enhance the manuscript’s currency.

Figures and Tables: Separate ROC curve figures have been retained, and a combined ROC curve (Fig. 7) has been added. This recommendation has thus been addressed. Table captions have been made more technical; however, for Table 3 (different threshold values), the explanatory notes remain brief. A more detailed interpretation of the results would be beneficial.

Experimental design

Methodological Clarity: The authors have clearly described the NLP process (VADER, TF-IDF, GloVe embeddings) and the libraries used. This addresses the earlier critique. Nevertheless, the rationale for excluding certain variables (e.g., economic/military categories) should be more thoroughly justified.

Data Cleaning: The rationale for excluding 60% of the data (short-lived or incomplete topics) has been explained. While this is a reasonable approach, presenting a quantitative assessment of its impact on validity (e.g., distribution/characteristics of excluded versus retained data) would make the justification more convincing.

Model Selection: The justification for selecting traditional ML models (RF, XGBoost, LightGBM, CatBoost, SVM) has been explicitly stated, and the decision to leave transformer-based models for future work is reasonable. While this explanation is satisfactory, the inclusion of a small-scale comparative experiment (e.g., a basic LSTM benchmark) would have significantly strengthened the manuscript. Its absence remains a methodological limitation.

Hyperparameters: Grid and random search strategies, along with parameter values, have been reported. This adequately addresses the previous concerns regarding transparency.

Validity of the findings

Class Imbalance: The authors report using SMOTE in combination with class weighting, and they have revised the manuscript to include performance comparisons across different thresholds. This largely addresses the original critique. However: The potential risk of overfitting caused by synthetic data after SMOTE should be emphasized more strongly in the discussion. Although confusion matrices have been added, the distribution of errors within the “popular” class requires more detailed commentary. At present, this analysis remains primarily at the visual level.

Generalizability: The limitation of relying solely on Facebook data has been more explicitly acknowledged. Nevertheless, it is still recommended that the authors specify a more concrete plan for future work, such as a small-scale cross-platform validation (e.g., testing on a Twitter subset).

Additional comments

The manuscript has been substantially improved compared to the initial version. The most critical issues (language clarity, methodological transparency, limited literature review, insufficient metrics) have largely been addressed. However, three aspects still require further development:

1. A quantitative analysis of excluded data (types of topics omitted and how they differ from the retained sample).

2. A more detailed interpretation of errors in the “popular” class.

3. Stronger support from recent and more diverse references.

These points remain essential for enhancing the overall robustness and impact of the study.

·

Basic reporting

The authors have addressed all my comments.

Experimental design

Good

Validity of the findings

Good

Additional comments

None

Reviewer 3 ·

Basic reporting

Based on the detailed response letter and the tracked changes in the manuscript, the authors have sufficiently addressed the reviewer’s comments and performed the necessary revisions.

Experimental design

Based on the detailed response letter and the tracked changes in the manuscript, the authors have sufficiently addressed the reviewer’s comments and performed the necessary revisions.

Validity of the findings

Based on the detailed response letter and the tracked changes in the manuscript, the authors have sufficiently addressed the reviewer’s comments and performed the necessary revisions.

---

## Round 0.3 · accepted · Accept

· Academic Editor

Accept

Dear Authors,

All reviewers now think that you have carefully addressed all recommendations raised during the major revision. These improvements seem to considerably strengthen the manuscript’s scientific contribution and overall impact.

Best wishes,

Reviewer 1 ·

Basic reporting

The authors have made substantial improvements in language and expression during the revision. Long and complex sentences have been simplified, and the methodology section is now more fluent and accessible. The literature review has been updated with recent studies from 2023–2025, particularly strengthening the Facebook and cross-platform context . Table and figure captions have been expanded, with Table 3 including a more detailed interpretation of threshold effects. These revisions have significantly enhanced the overall reporting quality of the manuscript.

Experimental design

The authors have clarified methodological descriptions and provided justifications for variable selection. The rationale behind excluding 60% of the dataset has been explained in detail, and a comparative analysis between excluded and retained data has been added . The choice of machine learning algorithms has been explicitly justified, and hyperparameter settings are now transparently reported. Class imbalance has been addressed through SMOTE and class weighting, with additional discussion of the potential overfitting risks introduced by synthetic data. These refinements have improved the rigor and transparency of the experimental design.

Validity of the findings

A quantitative analysis of excluded versus retained data has been incorporated, strengthening the validity of the data cleaning process. The error distribution in the “popular” class, particularly the tendency toward false negatives, has been discussed in greater detail. The conclusion section explicitly acknowledges the study’s limitations, including the absence of deep learning benchmarks and the reliance on a single platform, while also outlining concrete plans for cross-platform validation in future work. Collectively, these additions enhance both the internal validity and the generalizability of the findings.